# A Comprehensive Review of Immunotherapy Clinical Trials for Metastatic Urothelial Carcinoma: Immune Checkpoint Inhibitors Alone or in Combination, Novel Antibodies, Cellular Therapies, and Vaccines

**DOI:** 10.3390/cancers16020335

**Published:** 2024-01-12

**Authors:** Dixita M. Patel, Ruba Mateen, Noor Qaddour, Alessandra Carrillo, Claire Verschraegen, Yuanquan Yang, Zihai Li, Debasish Sundi, Amir Mortazavi, Katharine A. Collier

**Affiliations:** 1Division of Medical Oncology, Department of Internal Medicine, College of Medicine, The Ohio State University, Columbus, OH 43210, USA; 2Department of Internal Medicine, Franciscan Health Olympia Fields, Olympia Fields, IL 60461, USA; 3Department of Internal Medicine, Advocate Christ Medical Center, Oak Lawn, IL 60453, USA; 4The Ohio State University Comprehensive Cancer Center, Columbus, OH 43210, USA; 5Pelotonia Institute for Immuno-Oncology, The Ohio State University Comprehensive Cancer Center, Columbus, OH 43210, USA; 6Department of Urology, College of Medicine, The Ohio State University, Columbus, OH 43210, USA

**Keywords:** immunotherapy, urothelial carcinoma, bladder cancer, bispecific antibodies, cellular therapy

## Abstract

**Simple Summary:**

Metastatic urothelial carcinoma is sensitive to immunomodulation. Immune checkpoint inhibitors (ICIs) have been FDA approved for use as single agents to treat locally advanced or metastatic urothelial carcinoma (mUC) since 2016. Immunotherapy has a lower incidence of side effects and longer durability of response compared to chemotherapy. Most recently, both the first-line combinations of pembrolizumab plus enfortumab vedotin and of nivolumab plus gemcitabine plus cisplatin showed significant overall survival benefit over prior standard-of-care chemotherapy and will be the fifth and sixth new drug regimens approved for mUC in the past four years. Treatment options for mUC are expected to continue to rapidly evolve. Here, we summarize clinical trials of immunotherapy that have led to the current standard of care for mUC, then review clinical trials testing novel immunotherapeutic approaches. A comprehensive understanding of current clinical trials will enable anticipation of upcoming developments and future research directions for mUC.

**Abstract:**

Urothelial cancer is an immune-responsive cancer, but only a subset of patients benefits from immune checkpoint inhibition. Currently, single-agent immune checkpoint inhibitors (ICIs) and the combination of pembrolizumab with the antibody–drug conjugate enfortumab vedotin are approved to treat patients with metastatic UC (mUC). Approval of first-line nivolumab in combination with gemcitabine and cisplatin is expected imminently. Many treatment approaches are being investigated to better harness the immune system to fight mUC. In this review, we summarize the landmark clinical trials of ICIs that led to their incorporation into the current standard of care for mUC. We further discuss recent and ongoing clinical trials in mUC, which are investigating ICIs in combination with other agents, including chemotherapy, antibody–drug conjugates, tyrosine kinase inhibitors, and novel antibodies. Lastly, we review novel approaches utilizing bispecific antibodies, cellular therapies, and vaccines. The landscape of immunotherapy for mUC is rapidly evolving and will hopefully lead to better outcomes for patients.

## 1. Introduction

Bladder cancer alone has an estimated annual incidence in the US in 2023 of 82,290 cases causing 16,710 deaths, making it the fourth most common cancer in men [1,2]. Urothelial carcinoma (UC) includes cancers of the bladder, ureter, renal pelvis, and urethra. Metastatic UC (mUC) is aggressive, and outcomes are poor, but UC is an immune-responsive cancer, which holds great promise. The activity of intravesical Bacillus Calmette–Guerin (BCG) for non-muscle-invasive UC, first shown in 1976, was the earliest evidence of the immune responsiveness of UC [3]. In more recent years, immune checkpoint inhibitors (ICIs) have been used to harness the immune system to fight UC and have shown activity in the non-muscle-invasive, peri-operative muscle-invasive, and metastatic settings [4,5,6,7,8]. The currently approved ICIs are monoclonal antibodies to PD-1 or PD-L1, which interfere with immune-inhibitory ligand binding, in turn disinhibiting the anti-cancer immune response. 

The standard of care for mUC is rapidly evolving. With the October 2023 report of the results of the EV-302 trial, the treatment algorithm and landscape for mUC has dramatically changed. The previous standard of care for patients that were eligible to receive cisplatin was combination cisplatin-containing chemotherapy (gemcitabine plus cisplatin or dose-dense MVAC [methotrexate, vinblastine, doxorubicin, cisplatin]), then maintenance with the anti-PDL1 ICI avelumab [7,9,10,11,12]. For patients unable to receive cisplatin, either gemcitabine plus carboplatin followed by maintenance avelumab or the antibody–drug conjugate enfortumab vedotin-eivf (EV) in combination with the anti-PD1 ICI pembrolizumab were the standard-of-care first-line treatment options [7,9,10,11,12,13,14]. If a patient is not a platinum chemotherapy candidate (neither cisplatin nor carboplatin eligible), then pembrolizumab is approved in the first-line metastatic setting [15,16,17,18]. Next-line options include a single-agent anti-PD-1 or PD-L1 ICI for patients who have not previously received an ICI, or, if a susceptible *FGFR2* or *FGFR3* alteration is present, the oral FGFR inhibitor erdafitinib [8,19,20,21,22]. Subsequent options for patients that previously received platinum-containing chemotherapy and an ICI include the antibody–drug conjugates EV and sacituzumab govitecan-hziy (SG) [23,24]. Late-line options include single-agent taxane or pemetrexed chemotherapy [25,26,27].

As of October 2023, the new first-line standard of care is pembrolizumab plus EV for all patients with mUC. Pembrolizumab plus EV showed a median overall survival (mOS) of 31.5 months, which is notably longer than the mOS of 16.1–21.4 months with combination platinum chemotherapy with or without maintenance avelumab [7,9,28]. In certain situations, the anti-PD1 ICI nivolumab plus gemcitabine plus cisplatin may alternatively be used first-line [29]. However, until we have predictive biomarkers, we expect that in clinical practice most patients will receive pembrolizumab plus EV first-line. Going forward, first-line single-agent pembrolizumab will be reserved for patients that are expected to not tolerate concurrent EV. The optimal order of subsequent lines of therapy after pembrolizumab plus EV are unclear, but many patients may receive platinum-based chemotherapy next. The role of maintenance avelumab after second-line platinum or later-line single-agent ICIs in the context of previous front-line pembrolizumab plus EV is unclear. Subsequent treatment options include erdafitinib, sacituzumab govitecan, and late-line single-agent cytotoxic chemotherapy. 

In this review, we will focus on currently approved immunotherapy-containing regimens and ongoing clinical trials that are expected to influence the future of immunotherapy for mUC. Trials involving immunotherapy for non-muscle-invasive and localized muscle-invasive urothelial carcinoma are outside of the scope of this review. We first summarize the clinical trials leading to current approvals of ICIs in mUC, then discuss novel ICIs and trials of ICI containing maintenance regimens after front-line platinum chemotherapy. Next, we review combination regimens of ICIs with chemotherapy, antibody–drug conjugates (ADCs), tyrosine kinase inhibitors (TKIs) including FGFR inhibitors, and novel antibodies. Finally, we summarize studies of bispecific antibodies, cellular therapies, and neoantigen vaccines relevant to mUC. 

## 2. The Current Role of Immunotherapy in the Standard of Care for Metastatic Urothelial Carcinoma

To date, the only class of immunotherapeutic agents approved for the treatment of mUC are anti-PD-1/PD-L1 ICIs. As of November 2023, ICIs are FDA-approved as single agents (1.) for maintenance after not progressing on front-line platinum-containing chemotherapy, (2.) in the front-line metastatic setting for patients that are ineligible for any platinum chemotherapy (neither cisplatin nor carboplatin), and (3.) in the second-line metastatic setting. Most recently, pembrolizumab received FDA approval in combination with enfortumab vedotin for patients with mUC that are ineligible for cisplatin chemotherapy, which was expanded to include all patients with mUC based on the results of the practice-changing EV-302 trial reported in October 2023. We also anticipate FDA approval soon of first-line nivolumab plus gemcitabine and cisplatin for patients eligible to receive cisplatin based on CheckMate 901. All current ICI approvals in mUC are independent of PD-1/PD-L1 expression and are expected to remain so.

### 2.1. Pembrolizumab plus Enfortumab Vedotin

Enfortumab vedotin-ejfv (EV) is an ADC of a human IgG1 kappa monoclonal antibody directed against nectin-4, a cell-surface adhesion protein, conjugated to the microtubule-disrupting agent monomethyl auristatin E (MMAE). Single-agent EV received regular FDA approval for mUC after an ICI and platinum-based chemotherapy, or if cisplatin-ineligible, at least one prior ICI, based on an overall survival (OS) benefit compared to single-agent chemotherapy (hazard ratio [HR] 0.70, *p* = 0.001) in the phase 3 EV-301 trial regardless of nectin-4 expression [24,30].

In April 2023, EV plus pembrolizumab received accelerated approval by the FDA for mUC in cisplatin-ineligible patients, based on cohorts A and K of the EV-103 trial. The FDA approval does not specify the line of therapy, though it was based upon data in the front line. EV-103 is a multicohort phase 1/2 trial of EV alone and in combination with other agents in urothelial carcinoma. Cohorts A–G and K include patients with mUC, of which cohorts A, B, G, and K include EV in combination with pembrolizumab. Cohort A enrolled 45 cisplatin-ineligible patients with mUC to receive first-line EV plus pembrolizumab. The initial results showed that the ORR was 73.3%, complete response (CR) rate was 15.6%, and mOS was 26.1 months [14]. With further follow-up, median progression-free survival (mPFS) was 12.7 months (95%CI 6.11-NR), and mOS was 26.1 months (95%CI 15.51-NR) [31]. Building on these results, Cohort K was a randomized study of front-line EV +/− pembrolizumab that enrolled 149 cisplatin-ineligible patients with mUC. The ORR was 64.5% for EV + pembrolizumab with a 10.5% CR rate compared to an ORR 45.2% with a 4.1% CR rate for EV alone [32,33]. According to the pooled results of cohorts A and K reviewed by the FDA, among 121 patients who received EV plus pembrolizumab, the ORR was 68% (95%CI 59–76) and 12% CR rate [34]. Progression-free survival (PFS) and OS data are not yet mature. Given the higher reported ORR of pembrolizumab plus EV compared to the historical ORR of 41.2% with gemcitabine plus carboplatin, we suspect that in clinical practice pembrolizumab plus EV has been favored first line for cisplatin-ineligible patients over gemcitabine plus carboplatin [13].

The results of the phase 3 EV-302 trial were reported in October 2023 and are practice-changing. The trial randomized 886 patients 1:1 to first-line EV plus pembrolizumab or to gemcitabine plus platinum (cisplatin or carboplatin) [28]. The co-primary endpoints of PFS and OS were met: PFS HR 0.45 (95%CI 0.38–0.54, *p* < 0.00001) and OS HR 0.47 (95%CI 0.38–0.58, *p* < 0.00001). The median PFS was 12.5 months vs. 6.3 months, and median OS was 31.5 vs. 16.1 months for pembrolizumab plus EV vs. chemotherapy, respectively. The ORR was 67.7% with pembrolizumab plus EV compared to 44.4% with chemotherapy. Pembrolizumab plus enfortumab is the most effective regimen ever tested for mUC, was FDA approved in December 2023, and has become the new front-line standard of care for all patients. Now that patients will be receiving pembrolizumab plus EV first-line, the entire treatment paradigm for mUC will be altered, and the field is left with many questions about the sequencing of next-line therapies, as well as the impact on ongoing clinical trials that were designed based on the previous standard of care of frontline combination platinum-based chemotherapy followed by maintenance avelumab. Please note that the trials described throughout the remainder of this review were all designed in the pre-EV-302 era.

### 2.2. Nivolumab plus Gemcitabine plus Cisplatin in the First-Line Metastatic Setting

CheckMate-901 (NCT03036098) was a phase 3 trial that randomized 608 patients with mUC to six cycles of first-line gemcitabine plus cisplatin with or without nivolumab [29]. The results were presented in October 2023 concurrently with the above results of the EV-302 trial [35]. The addition of nivolumab resulted in significantly longer PFS (HR 0.72, 95%CI 0.59–0.88, *p* = 0.001) and OS (HR 0.78, 95%CI 0.63–0.96, *p* = 0.02). Median PFS was 7.9 vs. 7.6 months, median OS was 21.7 vs. 18.9 months, and ORR was 57.6% vs. 43.1% with nivolumab vs. without. Thus, the addition of nivolumab significantly improved outcomes compared to combination cisplatin-based chemotherapy. Based on the trial meeting its primary endpoint and showing an OS benefit, FDA approval of first-line nivolumab in combination with gemcitabine and cisplatin for mUC is expected. Of note, however, is that only 14.5% of patients in the gemcitabine plus cisplatin group received subsequent avelumab or pembrolizumab, so it is unclear if the experimental arm would have shown improved outcomes compared to platinum-based chemotherapy followed by maintenance avelumab. In clinical practice, while cross-trial comparisons should be avoided, we suspect that most patients will receive pembrolizumab plus EV first-line, rather than nivolumab plus gemcitabine plus cisplatin. It is unclear what the role is for this regimen after pembrolizumab plus EV. As noted below, there have been several unsuccessful attempts at combining cisplatin-based combination chemotherapy with an ICI in the first-line mUC setting, which may also temper excitement for the uptake of nivolumab plus gemcitabine plus cisplatin in practice.

### 2.3. Maintenance Avelumab after First-Line Platinum Chemotherapy

Avelumab is FDA-approved for maintenance treatment in patients that did not progress on first-line platinum-containing chemotherapy. The phase 3 JAVELIN Bladder 100 trial randomized 700 patients with unresectable locally advanced or metastatic UC to maintenance avelumab versus best supportive care after achieving an objective response or stable disease with four–six cycles of first-line gemcitabine plus cisplatin or carboplatin [7,9]. Patients that had progression on chemotherapy were excluded. The trial met the primary endpoint of OS benefit in favor of avelumab, with an HR for death of 0.69 (95% CI 0.56–0.86, *p* = 0.001) and a difference in mOS of 21.4 vs. 14.3 months. The benefit was seen regardless of PD-L1 status. Of note, a randomized phase II trial of maintenance pembrolizumab vs. placebo showed a significant PFS benefit (HR 0.65, log-rank *p* = 0.04), which was the primary endpoint, but no OS benefit (NCT02500121). Maintenance pembrolizumab has not received FDA approval [36]. Given the results of EV-302 and CheckMate 901, most, if not all, patients will soon be receiving an ICI in combination in the front-line setting with either pembrolizumab plus EV or nivolumab plus gemcitabine plus cisplatin; thus, the role for maintenance avelumab going forward is unclear.

### 2.4. Pembrolizumab in the First-Line Metastatic Setting for Platinum Chemotherapy Ineligible

In the front-line metastatic setting, only pembrolizumab has regular full FDA approval for patients that are deemed ineligible for any front-line platinum-based chemotherapy (neither cisplatin nor carboplatin containing) regimen, regardless of PD-1/PD-L1 status. The accelerated approval was initially justified by the multicenter, single-arm, phase II KEYNOTE-052 study that enrolled patients ineligible for cisplatin-based chemotherapy [18]. The objective response rate (ORR) was 24% (95% CI 20–29%). Median OS was 11.3 months (95% CI 9.7–13.1) [17]. Unfortunately, the confirmatory phase 3 KEYNOTE-361 trial of pembrolizumab with or without chemotherapy compared to chemotherapy did not find an OS benefit of pembrolizumab alone compared to chemotherapy (HR 0.92 (95% CI 0.77–1.11)) [16], nor was there a statistically significant PFS or OS benefit from the addition of pembrolizumab to chemotherapy compared to chemotherapy alone. However, the FDA ultimately decided to give full approval to first-line single-agent pembrolizumab in patients ineligible for any chemotherapy. While this is a small population of patients, the FDA felt that it would be important to ensure that they continue to have an available treatment option [37,38]. In the era of front-line pembrolizumab plus EV, it is expected that a small population of patients that are ineligible for any platinum chemotherapy or EV, perhaps due to severe neuropathy or poor performance status, will continue to receive front-line single-agent pembrolizumab.

### 2.5. Single-Agent ICIs in the Second-Line Metastatic Setting

There are currently three ICIs (pembrolizumab, nivolumab, and avelumab) that are approved in the second-line metastatic setting after prior platinum chemotherapy, regardless of PD1/PD-L1 status. Of these, an OS benefit has only been reported with pembrolizumab; thus, it has been favored in clinical practice. Of note, the anti-PDL1 antibodies atezolizumab [39] and durvalumab [40,41] previously received accelerated approval in this setting; however, they failed to yield an OS benefit compared to chemotherapy in phase 3 trials [42,43,44,45] and were voluntarily withdrawn. The role for any single-agent ICI in the second-line or beyond is unclear in the era of front-line pembrolizumab plus EV.

#### 2.5.1. Pembrolizumab

Pembrolizumab was approved for second-line metastatic UC based on the phase III KEYNOTE 045 study [8]. On this trial, patients with confirmed UC that progressed after platinum-based chemotherapy received either pembrolizumab 200 mg every 3 weeks or chemotherapy (paclitaxel, docetaxel, or vinflunine). OS was higher with pembrolizumab compared to chemotherapy (HR 0.73, 95% CI 0.59–0.91, *p* = 0.002; mOS 10.3 vs. 7.4 months). Additionally, the ORR was significantly higher in the pembrolizumab group (21.1% vs. 11.4%, *p* = 0.001). There was no difference in PFS. The pembrolizumab group had lower rates of treatment-related adverse events (60.9% vs. 90.2%). This study is the only to date that has reported an OS benefit of an ICI over standard chemotherapy in patients with platinum-refractory UC and, thus, has been the preferred single-agent ICI for this setting in clinical practice.

#### 2.5.2. Nivolumab

The phase II, single-arm CheckMate 275 trial tested nivolumab 3 mg/kg every 2 weeks in 270 patients with metastatic or surgically unresectable UC who had previously received platinum-based chemotherapy [20]. The ORR, which was the primary endpoint, was 19.6% (95% CI 15.0–24.9%). Eighteen percent had a grade 3–4 treatment-related adverse event. Based on this clinical benefit and the reasonable safety profile, nivolumab received accelerated approval regardless of PD-L1 status. A confirmatory larger trial is needed but not expected. Given the lack of phase 3 or OS data, nivolumab has been uncommonly used in this setting in clinical practice.

#### 2.5.3. Avelumab

Avelumab was evaluated in the phase 1b dose-expansion JAVELIN Solid Tumor trial [46,47]. A pooled analysis of two of the cohorts included 161 evaluable patients with mUC who received avelumab at progression after previous platinum chemotherapy and showed an ORR of 16.5% (95% CI 11.9–22.4%), a 4.1% CR rate, and a 12.4% partial response (PR) rate. Twenty-nine percent of patients had an infusion reaction, and eight percent experienced a treatment-related serious adverse event including one treatment-related death from pneumonitis. Based on this trial, avelumab received accelerated approval from the FDA in 2017 for use in advanced UC after progression on platinum-based chemotherapy. Of note, a larger confirmatory trial is needed, and avelumab is rarely used in this setting in clinical practice due to the lack of OS or phase 3 data.

## 3. Investigational Approaches Involving Immunotherapy for Metastatic Urothelial Carcinoma: Recent and Ongoing Clinical Trials

### 3.1. Novel Anti-PD-1, Anti-PD-L1, and Anti-CTLA-4 ICIs

Several novel ICIs targeted to PD-1, PD-L1, and CTLA-4 are in development as single agents for metastatic cancers, including in various lines of treatment for mUC, as shown in Table 1.

Novel anti-PD-1 antibodies in clinical trials for mUC include retifanlimab and a subcutaneous formulation of sasanlimab. A phase 2 POD1UM-203 trial of retifanlimab in metastatic UC, non-small-cell lung cancer (NSCLC), melanoma, and renal cell carcinoma (RCC) showed antitumor activity similar to other anti-PD-1 ICIs, with an ORR of 37.9%, disease control rate of 55.2%, mPFS of 5.7 months, and mOS of 15.2 months among the 29 patients with mUC [48]. The phase 1 dose escalation and expansion trial of IV and SQ sasanlimab in metastatic solid tumors included 39 patients with mUC. No DLTs were experienced, and they concluded that sasanlimab is safe and tolerable [49]. Among the 38 evaluable patients with mUC who received SQ sasanlimab, the ORR was 18.4% (95%CI 7.7–34.3%), mPFS 2.9 months (95%CI 1.9–3.8) by RECIST v1.1, and mOS 10.9 months (95%CI 7.2–19.9) [50]. Sasanlimab is being further developed via an ongoing phase 3 trial with and without BCG for non-muscle-invasive bladder cancer (NCT04165317), a phase 2 trial with stereotactic body radiotherapy for muscle-invasive bladder cancer (NCT05241340), and a phase 1 trial in combination with an anti-TIGIT Ab for solid tumors including mUC (NCT04254107), but no current dedicated trials for mUC.

There are four other anti-PD-1/PD-L1 antibodies being studied in phase 2 trials in China for mUC: tislelizumab, rulonilimab, toripalimab, and socazolimab. Tislelizumab given to 113 patients with mUC after platinum chemotherapy demonstrated an ORR of 24% (95%CI 16–33), CR rate of 8.8%, mPFS of 2.1 months, and mOS of 9.8 months [51]. Based on this study, tislelizumab was approved in China for mUC and is being investigated for high-risk non-muscle-invasive bladder cancer in combination with nab-paclitaxel (NCT04730219) [52]. Toripalimab is also approved in China for mUC after platinum-containing chemotherapy based on a phase 2 trial of 151 patients with mUC that showed an ORR of 25.8%, mPFS of 2.3 months (95%CI 1.8–3.6, and mOS of 14.4 months (95%CI 9.3–23.1) [53,54]. The status and results of the trials of rulonilimab and socazolimab for mUC are unknown.

Ipilimumab was the first approved anti-CTLA-4 antibody for cancer, though it does not carry an approval for UC. A phase 2 trial of nivolumab with and without ipilimumab after platinum-based chemotherapy tested two dosing regimens of ipilimumab plus nivolumab. The highest ORR was 38% in the combination arm of ipilimumab 3 mg/kg with nivolumab 1 mg/kg but with more toxicity [55]. Subsequently, the phase 3 CheckMate-901 trial of first-line ipilimumab plus nivolumab compared to chemotherapy for patients whose tumor cells express ≥1% did not show an OS benefit (NCT03036098) [56,57]. However, tremelimumab is another anti-CTLA-4 antibody being developed for the treatment of several solid tumors. A phase 2 trial is testing the activity of single-agent tremelimumab in mUC after prior PD-1/PD-L1 blockade. The results have not yet been reported. Several studies involving tremelimumab in combination with chemotherapy and anti-body drug conjugates for mUC are also ongoing and summarized in other sections below.

### 3.2. Maintenance ICI-Containing Combination Regimens

Since maintenance avelumab became the standard of care after front-line platinum-containing chemotherapy, multiple trials were initiated looking at novel immunotherapy combination regimens for maintenance, as summarized in Table 2. All are ongoing, and none have reported final results. Since pembrolizumab plus EV has been approved in the front-line setting for mUC, it is not clear what impact the results of these ongoing maintenance ICI regimens will have unless they are able to challenge the OS benefit seen in EV-302. Further, it will be difficult to accrue to these trials in parts of the world with access to pembrolizumab plus EV; thus, it may not be possible to complete these trials.

There are several ongoing single-arm phase 2 trials of avelumab in combination with another agent: talazoparib, lurbinectedin, or copanlisib [58,59,60]. The trial of avelumab with MRx0518, a proprietary Enterococcus gallinarum strain, was withdrawn (NCT05107427) [61]. There is also a single-arm phase 2 trial of maintenance ipilimumab plus nivolumab. PFS is the primary outcome of all these studies.

There are three ongoing randomized phase 2 trials of combination maintenance immunotherapy regimens. First, the PRESERVE 3 trial is a randomized phase 2 trial of CDK4/6 inhibitor trilaciclib plus chemo followed by trilaciclib plus avelumab maintenance compared to platinum chemo followed by avelumab maintenance [60]. A press release has reported that 94 patients were enrolled, and the ORR is similar between arms: ORR 40.0% in the trilaciclib arm vs. 46.7% in the control arm [62]. Follow-up is ongoing for PFS (primary endpoint), safety, and other secondary efficacy endpoints. Second, the JAVELIN Bladder Medley study is an ongoing phase 2 randomized, parallel four-arm, umbrella study of maintenance avelumab with or without either sacituzumab govitecan (SG), the anti-TIGIT antibody M6223, or the IL-15 receptor agonist NKTR-255. Third, the TROPHY U-01 trial is a phase 2 trial with multiple arms and cohorts, most of which include SG, two of which are in the maintenance space. Cohort 4 randomizes to first-line cisplatin plus sacituzumab govitecan for up to six cycles, followed by maintenance with sacituzumab govitecan and either zimberelimab (anti-PD1) or avelumab. Cohort 5 enrolls patients after four–six cycles of first-line gemcitabine plus cisplatin and randomizes them to three arms of zimberelimab +/− sacituzumab govitecan vs. avelumab [63]. The results of neither cohort have yet been reported.

The randomized MAIN-CAV trial of maintenance avelumab +/− cabozantinib with a primary endpoint of OS is the only phase 3 in this space and is actively recruiting [64].

### 3.3. Combination ICI Regimens for Metastatic Urothelial Carcinoma

#### 3.3.1. Cytotoxic Chemotherapy in Combination with ICIs

Clinical trials investigating chemotherapy with checkpoint inhibitors in mUC are summarized in Table 3.

With the exception of nivolumab plus gemcitabine plus cisplatin in CheckMate-901 (mentioned above in Section 2.2), results to date from trials combining ICIs with cytotoxic chemotherapy for mUC in the front-line setting have shown no significant benefit. Two of the four large phase 3 trials evaluating the benefit of adding an ICI to platinum-based chemotherapy failed to meet their primary endpoint. KEYNOTE-361 randomized 1010 patients 1:1:1 to pembrolizumab, gemcitabine/platinum chemotherapy, or pembrolizumab plus gemcitabine/platinum chemotherapy. The dual primary endpoint of superiority with regard to PFS and OS in the pembrolizumab plus chemotherapy versus chemotherapy alone groups did not meet the prespecified significance thresholds [16]. In other words, there was no proven benefit with the addition of pembrolizumab to platinum-based chemotherapy.

IMvigor130 enrolled 1213 patients who received atezolizumab, gemcitabine/platinum chemotherapy, or atezolizumab plus gemcitabine/platinum chemotherapy [65]. Again, the dual primary endpoints were PFS and OS for chemotherapy +/− atezolizumab. In this case, PFS was statistically significantly better (HR 0.82, 95%CI 0.70–0.96, *p* = 0.007), but there was no difference in OS (HR 0.83, 95%CI 0.69–1.0, *p* = 0.027), even with longer follow-up [66,67]. A phase 2 trial studying the effect of dosing schedules of atezolizumab plus gemcitabine/chemotherapy on ORR has not yet been reported (NCT03093922). Early results from a single-arm phase 2 trial of atezolizumab with split-dose cisplatin and gemcitabine showed an ORR of 40.9% and mPFS of 6.9 months (95%CI 6.4–9.2) (NCT04602078) [68]. OS data are not yet mature.

A single-arm phase 2 trial of gemcitabine/cisplatin plus ipilimumab in mUC enrolled 36 patients and did not meet the primary endpoint of OS [69]. The ORR was 69%, with a higher response rate among patients with somatic DNA damage response mutations.

Finally, the phase NILE trial (NCT03682068) is ongoing and randomizing patients 1:1:1 to durvalumab plus chemotherapy, durvalumab plus tremelimumab plus chemotherapy, or chemotherapy alone [70]. The co-primary endpoints are PFS and OS for durvalumab plus chemotherapy versus chemotherapy, and also OS of each immunotherapy-containing arm versus the chemotherapy arm in patients with high PD-L1 expression.

The phase 2 trial of gemcitabine/cisplatin with or without avelumab was terminated after the negative results of KEYNOTE-361 were reported and avelumab was approved in the maintenance setting (NCT03324282).

There are three ongoing trials in China combining an ICI with chemotherapy in the front line that do not yet have results: a phase 3 trial of gemcitabine plus platinum chemotherapy with or without tislelizumab (NCT03967977), a phase 3 trial of gemcitabine/platinum chemotherapy with or without toripalimab (NCT04568304), and a single-arm phase 2 trial of toripalimab plus nab-paclitaxel with or without cisplatin (NCT04211012). The phase 1 trial (NCT04603846) of first-line socazolimab plus nab-paclitaxel for mUC enrolled 20 patients in China and reported an ORR 52.9% (95%CI 27.81–77.02) [71].

Two trials are specifically studying chemotherapy plus ICI for patients with mUC who are ineligible for cisplatin. The phase 2 INDUCOMAIN trial (NCT03390595), tested avelumab for 2 cycles, then combination gemcitabine/carboplatin with avelumab, then maintenance avelumab compared to gemcitabine plus carboplatin in the front-line setting for cisplatin-ineligible patients. There was no benefit in the avelumab arm [72]. A phase 2 trial of nivolumab with gemcitabine and either carboplatin or oxaliplatin has not yet reported results (NCT03451331).

Finally, there are several clinical trials of ICIs with chemotherapy in later lines. A phase 1b/2 trial of pembrolizumab with docetaxel or gemcitabine showed that the combinations are safe, but the results of the dose-expansion cohorts are awaited (NCT02437370) [73]. The phase 1b trial of avelumab plus docetaxel showed that the combination was safe. Among 20 evaluable patients, the ORR was 70%, with a 30% CR rate and mPFS of 9.2 months (NCT03575013) [74]. The single-arm phase 2 PEANUT study of pembrolizumab with nab-paclitaxel enrolled 70 patients and showed an ORR of 38.6% (95%CI 27–51), 14.3% CR rate, and mPFS of 5.9 months (95%CI 3.1–11.5) [75]. A single-arm phase 2 trial of pembrolizumab with paclitaxel enrolled 27 patients and found an ORR of 33% and mOS of 11.7 months (95% CI 8.7-NR) [76]. Results are awaited for the phase 2 trial of atezolizumab with or without eribulin (NCT03237780), a phase 2 trial of an anti-PD1 with liposomal doxorubicin (NCT04101812), a phase 2 trial of pemetrexed with avelumab in MTAP-deficient mUC (NCT03744793) [77,78], and a phase 2 trial of pemetrexed with zimberelimab and etrumadenant, an adenosine A2 receptor antagonist (NCT05335941).

Given the new treatment paradigm of front-line pembrolizumab plus EV, we suspect that if there is a future role for immunotherapy plus chemotherapy in mUC, it will be in the second line or beyond, and likely with a novel non-ICI immunotherapy, or in triplet combination.

#### 3.3.2. Antibody–Drug Conjugates in Combination with ICIs

Table 4 summarizes clinical trials combining ICIs with the antibody–drug conjugates (ADCs) directed at nectin-4, trop-2, or HER2. The mechanism of ADCs combined with ICIs is illustrated in Figure 1.

**Table 4 cancers-16-00335-t004:** Clinical trials of antibody–drug conjugates in combination with immune checkpoint inhibitors for metastatic urothelial carcinoma.

Regimen	Drug Classes	Inclusion Criteria	Trial	Phase	Status (March 2023)	Identifier	ORR	mPFS (Months)	mOS (Months)
EV + pembrolizumab	ADC + anti-PD1	Cohort B: 2L+ mUC	EV-103	1/2	Active, not recruiting	NCT03288545	NA	NA	NA
EV + pembrolizumab vs. chemo	ADC + anti-PD1 vs. chemo	1L mUC	EV-302	3	Recruiting	NCT04223856	67.7% vs. 44.4%	12.5 vs. 6.3	31.5 vs. 16.1
EV + platinum + pembrolizumab	ADC + chemo + anti-PD1	Cohort G: 1L mUC	EV-103	1/2	Active, not recruiting	NCT03288545	NA	NA	NA
EV + platinum + pembrolizumab	ADC + chemo + anti-PD1	Arm C: 1L mUC	EV-302	3	Recruiting	NCT04223856	NA	NA	NA
EV + sitravatinib + pembrolizumab	ADC + TKI + anti-PD1	Cohort 9: mUC after platinum and ICI		2	Terminated	NCT03606174	NA	NA	NA
EV + pembrolizumab	ADC + anti-PD1	Non-urothelial and variant histology mUC		2	Not yet recruiting	NCT05756569	NA	NA	NA
SG + pembrolizumab	ADC + anti-PD1	Cohort 3: mUC after platinum	TROPHY U-01	2	Recruiting	NCT03547973	41%	5.3	12.7
SG +/− zimberelimab +/− domvanalimab vs. gemcitabine + carboplatin	ADC +/− anti-PD1 +/− anti-TIGIT vs. chemo	Cohort 6: 1L mUC cisplatin-ineligible	TROPHY U-01	2	Recruiting	NCT03547973	NA	NA	NA
SG + ipilimumab + nivolumab	ADC + anti-CTLA-4 + anti-PD1	1L mUC cisplatin-ineligible		1/2	Active, not recruiting	NCT04863885	NA	NA	NA
T-DXd + nivolumab	ADC + anti-PD1	HER2 expressing mUC after platinum; breast cancer	DS8201-A-U105	1b	Unknown	NCT03523572	36.7%	6.9	11.0
DV +/− pembrolizumab	ADC + anti-PD1	HER2 expressing mUC	KEYNOTE-D78	2	Recruiting	NCT04879329	NA	NA	NA
DV + toripalimab	ADC + anti-PD1	mUC		1b/2	Unknown	NCT04264936	73.2%	9.2	63.2% at 2 years
DV + toripalimab vs. platinum chemo	ADC + anti-PD1 vs. chemo	1L HER2 expressing mUC		3	Recruiting	NCT05302284	NA	NA	NA

Abbreviations: ORR = objective response rate; mPFS = median progression-free survival; mOS = median overall survival; 1L = first line; 2L+ = second line or beyond; EV = enfortumab vedotin; SG = Sacituzumab govitecan; T-DXd = trastuzumab deruxtecan; DV = disitamab vedotin; ADC = antibody–drug conjugate; mUC = metastatic urothelial carcinoma; NA = not available.

**Figure 1 cancers-16-00335-f001:**
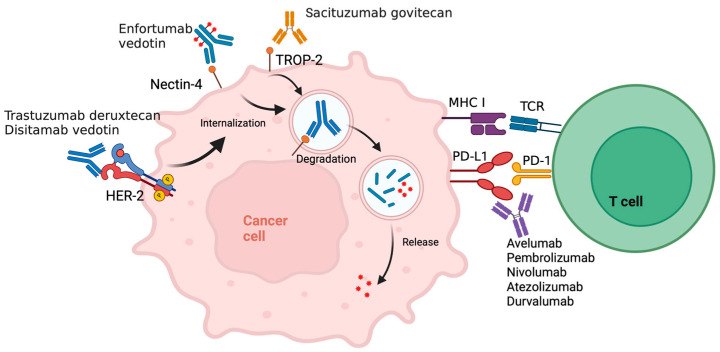
Mechanism of action of antibody drug conjugates (ADCs). The ADC (enfortumab vedotin, trastuzumab deruxtecan, disitamab vedotin, or sacituzumab govitecan) binds to their respective target on the cancer cell surface (nectin-4, HER2, or trop-2), leading to internalization, degradation, and release of the cytotoxic payload molecule (monomethyl auristatin E, topoisomerase I inhibitor, or SN-38) causing cell death. Concurrent administration of an anti-PD1/PDL1 antibody disinhibits T-cell-mediated tumor cell killing. Created with BioRender.

##### Enfortumab Vedotin plus ICI

As described above, pembrolizumab plus enfortumab vedotin recently received accelerated approval for mUC in patients that are ineligible for cisplatin, based on cohorts A and K of the EV-103 trial, and the phase 3 EV-302 trial showed superiority of EV plus pembrolizumab over platinum-containing chemotherapy in the front-line setting for all platinum-eligible patients. The other cohorts of EV-103 have not yet been reported, including cohort B testing EV plus pembrolizumab in the second line or beyond. Cohort G will test the triplet of EV plus platinum chemotherapy (cisplatin or carboplatin) plus pembrolizumab in the front line for mUC and will be further studied in EV-302 arm C. If the side-effect profile is tolerable, the triplet regimen has the potential to outperform the current standard of care.

A phase 2 multiple cohort trial also studied the triplet of EV plus pembrolizumab plus the multi-TKI sitravatinib in patients with mUC who have previously received platinum chemotherapy and an ICI. The trial was terminated by the sponsor after enrolling 260 patients into nine cohorts. Cohort 9 was a dose escalation at three dose levels for sitravatinib plus EV plus pembrolizumab in 16 patients with mUC that had previously received both ICI and platinum chemotherapy. The optimal dose for the combination regimen has not yet been reported. According to posted results, four of sixteen (25%) patients had an objective response. The other cohorts are discussed below in the next section.

A phase 2 trial is ongoing to study EV plus pembrolizumab in non-urothelial (adenocarcinoma, pure squamous) and pure variant histology UC (micropapillary, plasmacytoid, clear cell, etc.) cancers of the urinary tract. Small-cell carcinoma is excluded.

##### Sacituzumab Govitecan plus ICI

Sacituzumab govitecan-hziy (SG) is an ADC of an antibody to the cell-surface protein Trop-2 linked to the topoisomerase inhibitor and irinotecan metabolite SN-38. It received FDA accelerated approval as a single agent for patients with mUC who previously received platinum-containing chemotherapy and an ICI based on the single-arm cohort 1 of the phase 2 TROPHY U-01 trial, which showed an ORR of 27% (95%CI 19.5–36.6), mPFS of 5.4 months (95% CI 3.5–7.2), and mOS of 10.9 months (95%CI 9.0–13.8) [23]. Single-agent SG is a standard-of-care option in the third line setting for mUC.

Given the proven single-agent activity in mUC, there are ongoing efforts to combine SG with ICIs. The phase 2 TROPHY U-01 trial is a multicohort study looking at SG in combination with several other drugs for mUC. Cohort 3 includes patients with mUC that previously received platinum, who will receive SG in combination with pembrolizumab. This single-arm cohort of 41 patients showed an ORR of 41% (95% CI 26.3–57.9), mPFS of 5.3 months (95%CI 3.4–10.2), and mOS of 12.7 months (95%CI 10.7-NE) [79,80]. For cohort 6, patients with mUC that are ineligible for cisplatin in the first line will be randomized to SG alone, SG plus zimberelimab (anti-PD1), SG plus zimberelimab plus domvanalimab (anti-TIGIT), or gemcitabine plus carboplatin [81]. The results of this cohort have not yet been reported.

A phase 1/2 trial of first-line SG plus ipilimumab and nivolumab for four cycles followed by nivolumab maintenance in cisplatin-ineligible patients with mUC is also ongoing [82].

##### HER2 Antibody–Drug Conjugate plus ICI

There are two HER2-directed ADCs being investigated alone and in combination with ICIs for mUC: trastuzumab deruxtecan and disitamab vedotin. HER2 is expressed in 6.7–52.3% of mUC, but previous attempts at targeting HER2 with trastuzumab in mUC have been unsuccessful [83,84]. A single-arm phase 2 trial of trastuzumab added to paclitaxel, carboplatin, and gemcitabine (PCG) chemotherapy had a 57% confirmed ORR but had higher than expected cardiotoxicity and has not been compared to PCG chemotherapy in a randomized trial [83]. A phase 2 trial of gemcitabine plus platinum (cisplatin or carboplatin) with or without trastuzumab showed no PFS or ORR benefit with the addition of trastuzumab but was underpowered due to a lower-than-expected HER2 positivity rate among screened patients [85]. Despite previous unimpressive results with trastuzumab, a small phase 2 trial of afatinib showed activity in mUC with a HER2/3 alteration, and there is renewed interest in targeting HER2 in mUC with the advent of HER-directed ADCs, which are being studied in combination with ICIs in multiple clinical trials [86].

Trastuzumab deruxtecan (T-DXd) is a humanized antibody to HER2 conjugated to a topoisomerase I inhibitor. The phase 1b trial of T-DXd plus nivolumab consists of a dose escalation cohort followed by expansion cohorts in both HER2 expressing mUC and metastatic breast cancer. HER2 expression in mUC was defined as either IHC 2+ or 3+ (cohort 3) or IHC 1+ “HER2-low” (cohort 4) [87,88,89,90]. Among the 30 enrolled patients with mUC and HER2 IHC 2+/3+ (cohort 3), the ORR was 36.7% (95%CI 19.9–56.1), with a 13.3% CR rate, mPFS of 6.9 months (95%CI 2.7–14.4), and mOS of 11.0 months (95%CI 7.2-NR) [88]. The HER2-low cohort was closed due to poor accrual. Of note, drug-related interstitial lung disease (ILD) of any grade, a known toxicity of T-DXd, occurred in 23.5% of patients.

Disitamab vedotin (RC48-ADC) is an antibody to HER2 conjugated to MMAE. In a phase 2 single-arm study, disitamab vedotin alone showed activity in HER2 expressing (IHC 2+/3+) mUC after chemotherapy, with an ORR of 51.2% (95%CI 35.5–66.7%), mPFS of 6.9 months (95%CI 5.6–8.9), and mOS of 13.9 months (95%CI 9.1-NR) [91]. There is an ongoing phase 2 multicohort study of disitamab vedotin alone and in combination with pembrolizumab for HER2 expressing mUC. There are Five cohorts A–E, of which cohorts D and E will be enrolling only in Japan. Given the results of EV-302, cohorts A and B of single-agent disitamab vedotin in HER2-positive or -low mUC that require prior platinum and do not allow prior EV will not be able to accrue anywhere that has access to pembrolizumab plus EV. Cohorts C and E of disitamab vedotin with or without pembrolizumab in first-line HER2-positive mUC may also have difficulty accruing if investigators do not have equipoise regarding the effectiveness of this regimen relative to standard-of-care first-line pembrolizumab plus EV. Cohort D of single-agent disitamab vedotin after prior platinum, ICI, and EV, regardless of HER2 status, remains relevant and feasible. Further development of disitamab vedotin, which is a promising drug, may require redesign of this clinical trial to take into account the latest standard of care.

A phase 1b/2 trial of disitamab vedotin plus toripalimab (anti-PD1) for mUC is being conducted in China. Results from 41 enrolled patients with mUC showed an ORR of 73.2% (95%CI 57.1–85.8), with a 9.8% CR rate and mPFS of 9.2 months (95%CI 5.7–10.3) [92,93,94]. Sixty-one percent of the patients received disitamab vedotin plus toripalimab as first-line treatment, among whom the ORR was 76.0%. Only 59% of patients were IHC 2+/3+ positive for HER2, and even among patients with 0 HER2 expression by IHC, the ORR was 33.3%. OS data are not yet mature. There is an ongoing randomized phase 3 trial comparing disitamab vedotin plus toripalimab for front-line treatment of HER2 expressing (IHC 1+, 2+ or 3+) mUC compared to platinum-containing chemotherapy.

#### 3.3.3. Tyrosine Kinase Inhibitors plus ICIs

Clinical trials combining a tyrosine kinase inhibitor (TKI) with ICIs are summarized in Table 5, with a particular focus on FGFR inhibitors.

##### Fibroblast Growth Factor Receptor (FGFR) Inhibitors plus ICIs

Erdafitinib is the only currently approved TKI for mUC. It is an oral FGFR1-4 inhibitor and was approved as a single agent in the second-line-or-beyond metastatic setting after platinum-based chemotherapy for patients with cancers that harbor a susceptible *FGFR2* or *FGFR3* fusion or *FGFR3* mutation based on a 40% ORR in the BLC2001 study and a confirmed OS (HR 0.64, 95%CI 0.47–0.88, *p* = 0.005), PFS (HR 0.58, 95%CI 0.44–0.78, *p* = 0.0002), and ORR (46% vs. 12%) benefit compared to docetaxel or vinflunine in cohort 1 of the THOR study [19,21]. Cohort 2 of the THOR study showed no significant difference in either the primary endpoint of OS or PFS between erdafitinib and single-agent pembrolizumab but did show a higher ORR of 40.0% with erdafitinib compared to 21.6% with pembrolizumab [22].

The phase 1b/2 NORSE trial is studying erdafitinib plus cetrelimab (anti-PD1) with or without platinum chemotherapy in patients with mUC and an *FGFR2/3* gene alteration. In the phase 1b study of 22 patients, the combination of erdafitinib plus cetrelimab was concluded to be safe, and a recommended phase 2 dose was determined [95]. In the phase 2 portion, patients receive erdafitinib with or without cetrelimab in the front line and must be cisplatin-ineligible. The final results for the NORSE trial for cisplatin-ineligible patients in the front-line mUC setting showed an ORR of 54.5% (95%CI 38.8–69.6), with a 13.6% CR rate and mPFS of 10.97 months (95%CI 5.45–13.63) among the 44 patients in the erdafitinib plus cetrelimab cohort compared to an ORR of 44.2% (95%CI 29.1–60.1), with a 2.3% CR rate and mPFS of 5.62 months (95%CI 4.34–7.36) among the 43 patients in the erdafitinib-alone cohort [96,97].

Several other FGFR inhibitors are also being tested in clinical trials in combination with ICIs including futibatinib, rogaratinib, derazantinib, and LOXO-435. Futibatinib plus pembrolizumab is being studied in a phase 2 trial in platinum-ineligible cohorts with and without *FGFR3* mutations or *FGFR1-4* fusions/rearrangements. Preliminary safety results from the six patients included in the safety lead-in support ongoing study of this combination. Further results have not yet been reported [98].

The FORT-2 trial is a phase 1b/2 trial of rogaratinib, an oral pan-FGFR1-4 inhibitor, plus atezolizumab compared to placebo plus atezolizumab in patients with untreated cisplatin-ineligible mUC that have a tumor with high (3+ or 4+) FGFR1 or FGFR3 mRNA expression by RNAscope. The phase 1b portion of the study enrolled 26 patients. A maximum tolerated dose of rogaratinib in combination with atezolizumab was determined, and the combination is well tolerated. The ORR was 54%, with a CR rate of 13%. Among the 16 patients with low/negative PD-L1 protein and FGFR3 mRNA expression, the ORR rate was similar at 56% [99,100]. The authors concluded that further development of this combination is warranted.

The FIDES-2 trial is a multicohort phase1b/2 trial of derazantinib, an oral FGFR1-3 and CSF1R kinase inhibitor, with or without atezolizumab in patients with mUC and an FGFR1-3 mutation or fusion by next-generation sequencing. Cohorts 1 and 5 treat patients who received at least one prior line of therapy with derazantinib alone; cohort 2 treats patients with any advanced metastatic solid tumor with derazantinib plus atezolizumab; cohort 3 treats cisplatin-ineligible PD-L1-low mUC with first-line derazantinib plus atezolizumab; cohort 4 treats mUC resistant to prior FGFR inhibitor with derazantinib with or without atezolizumab [101]. The combination has been reported to be safe in 13 patients, and the recommended phase 2 dose was determined [102]. However, cohorts 1 and 5 did not show efficacy of derazantinib alone; the combined ORR was only 8.2% (95%CI 2.2–19.5), and PFS 0.3–8.8 months and 0.3–21.4 months [103]. Further development of derazantinib in mUC is unlikely.

There is a phase 1 trial of LOXO-435, a small molecule FGFR3 inhibitor, which has one cohort studying the combination of LOXO-435 with pembrolizumab in mUC in patients who have not previously received an FGFR inhibitor and who have a prespecified activating FGFR3 alteration [104]. Results are awaited.

A phase 2 trial of pemigatinib with and without pembrolizumab versus gemcitabine and carboplatin in front-line treatment for cisplatin-ineligible mUC with an FGFR mutation or rearrangement was terminated due to a business decision.

##### VEGFR and Multitargeted TKIs plus ICIs

Several vascular endothelial growth factor receptors (VEGFR) and multitargeted TKIs are being combined with ICIs for mUC, including cabozantinib, lenvatinib, sitravatinib, famitinib, and XL092.

Cabozantinib is a multitargeted TKI directed at c-Met, VEGFR2, AXL, FLT3, and RET, among others. It is approved for the treatment of advanced hepatocellular carcinoma (HCC) and metastatic renal cell carcinoma (RCC). A phase I basket trial of cabozantinib plus nivolumab with or without ipilimumab included patients with previously treated metastatic genitourinary (GU) cancers including mUC. Of 54 enrolled patients, the 15 with mUC experienced an ORR of 38.5% (95%CI 13.9–68.4), mPFS of 12.8 months (95%CI 6.9–18.8), and mOS of 25.4 months (95%CI 5.7–41.6) [105]. In the phase I expansion cohort of 30 patients with mUC who had received prior ICI, cabozantinib plus nivolumab resulted in an ORR of 16.0% with one CR and was concluded to be active and well tolerated, with evidence of immunomodulation on correlative analyses [106]. The ongoing phase 2 ICONIC trial is studying the combination of cabozantinib plus nivolumab and ipilimumab in patients with rare GU tumors, including histologic variants of bladder cancer and urothelial carcinoma metastatic to bone only. The single-arm phase 2 ARCADIA trial of cabozantinib plus durvalumab for mUC and advanced variant histology bladder cancer after previous platinum-containing chemotherapy enrolled 58 evaluable patients at interim analysis and showed an ORR of 39.7% (95%CI 27.1–53.4), with a 20% CR rate, an mPFS 7.6 of months, and OS of 11.6 months [107]. There is also an ongoing single-arm phase 2 trial of first-line cabozantinib plus pembrolizumab in cisplatin-ineligible patients with mUC.

Lenvatinib is a multi-TKI that targets VEGFR1-3, FGFR1-4, PDGFR, c-KIT, and RET, among others. It is approved for the treatment of advanced thyroid cancer, HCC, and RCC. The randomized phase 3 LEAP-011 compared front-line pembrolizumab with or without lenvatinib for the treatment of patients with mUC that are both cisplatin-ineligible and PD-L1-negative or platinum-ineligible regardless of PD-L1 expression. Unfortunately, this was a negative study. Among the 441 enrolled patients, there was no PFS (HR 0.91, 95%CI 0.71–1.16) or OS (HR 1.25, 95%CI 0.94–1.67) benefit with the addition of lenvatinib to pembrolizumab compared to pembrolizumab alone, and toxicities were also more frequent in the combination arm [108]. This combination regimen is not being further pursued for mUC.

Sitravatinib is a multi-TKI targeted at MET, Axl, VEGFR, PDGFR, KIT, FLT3, Trk, RET, DDR2, and others. It has been studied in a multicohort phase 2 trial in combination with nivolumab in various lines of therapy for mUC, and after different previous classes of treatment in both platinum-eligible and -ineligible patients [109]. Ultimately, the trial was terminated by the sponsor for strategic, not safety, reasons after enrolling 260 patients into nine cohorts. According to posted results, across cohorts the ORR ranged from 0 to 33%, with an mPFS of 3.5–7.8 months. ORR was highest in cohort 5, which included 53 patients with mUC that had previously received platinum chemotherapy but not an ICI. The ORR was 32.1%, with an mPFS of 3.9 months (95%CI 3.5–5.7) [110]. Cohort 2 of 23 patients that were ineligible for platinum but had previously received an ICI showed an ORR of 21.7% and the highest mPFS of 7.79 months (95%CI 3.94–14.62). Cohort 1 of 49 patients with mUC that had previously received platinum-based chemotherapy and an ICI had an ORR of only 12.2% [111]. Toxicities were manageable.

Famitinib is an oral multi-TKI to c-Kit, VEGFR2/3, PDGFR, and Flt1/3 being studied in combination with camrelizumab (anti-PD1) in a phase 2 multicohort adaptive non-randomized trial in patients with advanced UC, RCC, or gynecologic cancers in China. The cohort of patients with mUC after platinum chemotherapy enrolled 36 patients who experienced an ORR of 30.6% (95%CI 16.3–48.1%), mPFS of 4.1 months (95%CI 2.2–8.2), and mOS of 12.9 months (95%CI 8.8-NR) [112,113].

Finally, STELLAR-002 is a multicenter phase 1b basket trial in advanced solid tumors with two dose-expansion cohorts for mUC studying the multi-TKI XL092 alone or with nivolumab, nivolumab plus ipilimumab, or nivolumab plus relatlimab (anti-LAG3). XL092 targets MET, VEGFR2, AXL, and MER, among others. Patients with mUC are separated into cohorts based on whether or not they previously received an ICI, and they must not have received more than one–two lines of previous therapy [114]. Results are awaited.

To date, no TKI-plus-ICI regimen has reported responses that can challenge pembrolizumab plus EV in the front line, nor platinum chemotherapy. Front-line trials in cisplatin-ineligible patients are unlikely to be completed since pembrolizumab plus EV was approved. There may be a role for trials of TKIs in later lines or in triplet regimens, as results to date have shown some activity but without impressive or synergistic results.

#### 3.3.4. Novel Immunotherapeutic Antibodies plus ICIs

Novel antibodies are being tested in clinical trials in combination with ICIs for mUC (Table 6). Targets include OX40, GITR, α-GAL-9, GARP-TGFβ1, ILT3, LAIR1, and MIC-1. Most trials have not yet yielded results.

Four trials have enrolled patients with mUC to study agonist monoclonal antibodies to members of the tumor necrosis factor receptor superfamily expressed on immune cells, including OX40 and GITR, alone and in combination with ICIs. The first is a phase 1/2 trial of the anti-OX40 Ab INCAGN01949 with or without nivolumab/ipilimumab in advanced solid tumors. The phase 1 study, including 11 dose-escalation and safety-expansion cohorts, enrolled fifty-two patients but since only three patients experienced a response, the phase 2 dose expansion was not opened. The second was a phase 2 trial of atezolizumab with or without the anti-OX40 Ab vonlerolizumab (MOXR0916) in front-line treatment of mUC in patients ineligible for cisplatin. The study was terminated for slow accrual and business strategy after enrolling only five patients. The third study is a phase 1b/2 trial of the anti-OX40 Ab BGB-A445 with or without tislelizumab in the second–fourth-line metastatic setting for UC, RCC, or melanoma, which is ongoing. Finally, the anti-GITR agonist ICAGN01876 was studied in advanced solid malignancies in a phase 1/2 trial, with expansion cohorts in non-urothelial metastatic cancers. All but one patient experienced a treatment-emergent adverse event, and the ORR was 23.9% in head and neck squamous cell carcinoma, 16.7% in cervical cancer, and 0% in melanoma and gastric cancer. To date, clinical trials of TNF receptor agonists have been discouraging.

Gal-9, a ligand of the T-cell inhibitory receptor TIM-3, is being targeted in an ongoing phase 1/2 study of the anti-α-GAL-9 monoclonal antibody LYT-200 with or without tislelizumab or gemcitabine plus nab-paclitaxel in advanced solid malignancies, with a dose-expansion cohort for mUC. The results are awaited. Another phase 1 trial is studying a PD1 inhibitor (budigalimab) with a novel antibody to bind the GARP-latent TGFβ1 complex (ABBV-151) to prevent the release of active immunosuppressive TGFβ1 in advanced solid tumors including mUC [115]. Preliminary results showed no antitumor responses of the novel antibody as a single agent; however, in the mUC dose-expansion cohort in patients that had previously received an ICI, there were five responses among 36 patients (ORR 13.9%) with four more patients having stable disease for over 6 months [116]. Toxicities were manageable.

Three trials are studying novel antibodies designed to increase antitumor immunity by inhibiting immunosuppressive myeloid cells. First, NGM821 is an antagonist antibody designed to prevent ILT3 (LILRB4) on suppressive myeloid cells from interacting with fibronectin in the tumor microenvironment, a “myeloid checkpoint.” It is being given alone and in combination with pembrolizumab in advanced solid tumors including mUC. Similarly, there is a phase 1 trial of NGM438, an antagonist antibody to LAIR1, with or without pembrolizumab in advanced solid tumors. This molecule prevents the inhibitory LAIR1 receptor on myeloid cells from binding to collagen. Both trials are ongoing. Finally, the phase 1/2 first-in-human trial of the anti-macrophage inhibitory cytokine (MIC) -1 (anti-GDF15) Ab AZD8853 in patients with advanced mUC, NSCLC, or colorectal cancer was terminated “based on the overall risk-benefit profile observed to date” [117]. These attempts at targeting myeloid cells in mUC are early, and we expect novel therapeutics in this space in the future.

### 3.4. Bispecific Antibodies for Metastatic Urothelial Carcinoma

Bispecific antibodies (BsAbs) are linked antibodies to two different antigen targets, bringing them into proximity to bind and either activate or block/sequester nearby ligands. When one antibody binds to a T cell and one antibody is directed at a tumor cell, these are referred to as bispecific T-cell engagers (BiTEs) (Figure 2A). There are currently seven FDA-approved BsAbs for cancer treatment, four of which are BiTEs: blinatumomab targeting CD3 and CD19 for acute lymphoblastic leukemia, tebentafusp-tebn targeting CD3 and gp100 for uveal melanoma, mosunetuzumab-axgb targeting CD3 and CD20 for follicular lymphoma, epcoritamab-bysp and glofitamab-gxbm targeting CD3 and CD20 for B-cell lymphomas, teclistamab-cqyv targeting CD3 and BCMA for multiple myeloma, and amivantamab-vmjw targeting EGFR and MET for EGFR exon 20 insertion mutated non-small-cell lung cancer (NSCLC).

There is one ongoing trial of a BsAb specifically for mUC and a second trial of a BsAb only in metastatic non-prostate GU malignancies. The first is a randomized phase 2 trial of a BsAb to PD-1 and LAG-3 (RO7247669) with or without the anti-TIGIT antibody tiragolumab compared to single-agent atezolizumab in the first line for platinum-ineligible patients with mUC enrolling in China (NCT05645692) (Figure 2B). In the completed first-in-human phase I trial of RO7274669 in advanced solid tumors, there were grade 3 treatment-related AEs in 17.1% of patients, no grade 4–5 adverse events, and an ORR of 17.1% [118]. The second trial of a BsAb specific to non-prostate advanced GU malignancies combines bintrafusp alfa, designed to bind PDL1 while sequestering TGFβ, with NHS-IL12, an immuno-cytokine comprised of IL12 fused to an NHS76 antibody, designed to deliver IL12 to areas of tumor necrosis (NCT04235777) (Figure 2C) [119,120,121]. Additionally, patients may receive SBRT to a metastatic site. The trial is ongoing, but interim results showed no dose-limiting toxicity and an ORR across all arms of 36.4% with four of eleven patients responding [122]. Based on disappointing results of bintrafusp alfa in other solid tumors including NSCLC, further drug development by the company is in question.

There are many ongoing early-phase basket trials of BsAbs in advanced solid tumors that include mUC, some of which are listed in Appendix A. Several of these BsAbs simultaneously target two immune checkpoint molecules including PD-1, PDL-1, CTLA-4, LAG-3, TIGIT, TIM-3, or ICOS. Others are designed to sequester VEGF or TGFβ. Many are designed to bring immune cells in proximity with tumor cells by targeting both immune markers (CD137/4-1BB, CD27, PD-1, OX-40) and tumor markers (B7-H3, CD47, PD-L1, 5T4, ROR1, claudin, GPC3, HLA-G, PSCA). There are several bispecific antibodies being developed for HER2- or EGFR-overexpressing cancers, which occurs in up to 40% and 74% of mUC, respectively [84,123]. At least 11 trials are combining a BsAb with an ICI. Overall, there are limited reports of the activity of BsAbs in mUC, largely due to a lack of dedicated dose-expansion cohorts. One patient with mUC showed a partial response to GI-101, which is a CD80 x IL2R BsAb [124]. The field of BsAbs for solid tumors is still in its infancy and is primarily focused on proving safety to date. Significant progress is expected in the coming decade, particularly proving efficacy in specific solid tumor types, such as mUC.

### 3.5. Cellular Therapy for Metastatic Urothelial Carcinoma

Early clinical investigation of cellular therapies for mUC and metastatic solid tumors has been attempted. To date, most studies have used lymphocyte products. T cells can be obtained from the peripheral blood, or they can be tumor-infiltrating lymphocytes (TILs) isolated from the tumor tissue. Cell products can also be either autologous from the patient or allogeneic from a donor. Autologous products are often engineered and expanded ex vivo, then infused back into the patient. TCR-gene engineered lymphocytes are autologous T cells that are engineered ex vivo so that the T-cell receptor (TCR) recognizes an HLA-presented tumor antigen peptide. While chimeric antigen receptor T-cell therapies (CAR-T cells) can bind to any particular cancer cell surface antigen, engineered TCRs can only bind to MHC-presented peptides on the cancer cell surface (Figure 2D). There are currently six FDA-approved CAR-T therapies: four targeting CD19 in B-ALL, DLBCL, mantle cell lymphoma, or follicular lymphoma, and two targeting BCMA in multiple myeloma. To date, the only FDA-approved autologous cellular product for a solid tumor malignancy is sipuleucel-T for metastatic castration-resistant prostate cancer.

There are seven clinical studies involving cellular therapies specifically for the treatment of mUC (Table 7). Only one study has results; it included two patients with mUC who received autologous WT-1 directed dendritic cells plus chemotherapy. The regimen was safe, but neither patient had a response [125]. Three other studies have been completed or are ongoing, but none have posted, reported, or published results. A single-center phase 2 trial in Israel for mUC after platinum-based and ICI therapies is testing the safety and efficacy of autologous tumor-infiltrating lymphocytes (TILs) extracted from surgically resected bladder tumors (upper-tract UC excluded) and expanded ex vivo, then infused after lymphodepleting reduced intensity, non-myeloablative chemotherapy followed by high-dose IL2 up to 10 doses (NCT04383067). A phase 1/2 trial in China is testing the safety and efficacy of both PSMA-targeted and FRα-targeted fourth-generation CAR-T cells for mUC with no further treatment options (NCT03185468) (Figure 2D). An ongoing phase 1/2 trial in Belarus of autologous “cytokine-induced killer cells” for mUC or metastatic RCC is presumably testing a stimulated autologous NK cell product; however, little information is available (NCT05108077).

The remaining three studies of cellular therapies for mUC were terminated due to lack of funding or poor accrual. A single-center phase 2 trial in the US of autologous TILs extracted from the tumor and expanded ex vivo, then infused with pembrolizumab after lymphodepleting fludarabine + cyclophosphamide chemotherapy followed by alesleukin was withdrawn due to no accrual (NCT03935347). A phase 2 trial in China was initiated in patients with mUC who had previously received cisplatin chemotherapy to test the effect of autologous central memory T cells (NCT03389438). The study was terminated after enrolling six patients due to inadequate funding. A phase 1 trial in China attempted to give autologous T cells for mUC that were engineered ex vivo using CRISPR Cas9 to knock out *PDCD1*, the gene for PD-1 (NCT02863913). Lymphodepleting cyclophosphamide was to be given prior to the T cells, followed by IL-2. However, the trial was withdrawn prior to enrollment of any patients, citing lack of funding.

There are also several early-phase trials of cellular therapies in advanced and metastatic solid tumors including mUC, some which are listed in Appendix A. These include TCR-gene engineered lymphocytes to NY-ESO-1, MAGE-A1, or MAGE-A3; CAR-T cells to ROR2; donor lymphocytes; and autologous NK cells [126,127].

Overall, clinical trials of cellular therapies for mUC are very early and have not shown success, in part due to logistical and funding issues. There is significant further refinement and development needed in this space. Preclinical studies are exploring novel cellular targets as well as potentiating combination regimens [128,129,130].

### 3.6. Neoantigen Vaccines for Metastatic Urothelial Carcinoma

Vaccines for cancer therapy are designed to stimulate an immune response against tumor neoantigens, which are the consequence of somatic DNA mutations [131]. Vaccines studied in mUC have included inoculations with neoantigen peptides, viruses encoding target antigens or immunostimulatory molecules, or vesicles containing mRNA that encodes the neoantigen. Previous attempts at vaccine development for mUC have been safe but have shown little or no efficacy [132,133,134,135,136].

As summarized in Table 8, there are four recent or ongoing clinical trials of vaccines that are designed only for patients with mUC. A phase 2 trial gave CV301 with atezolizumab as the first-line treatment for patients with mUC who are not eligible for cisplatin (cohort 1) or after cisplatin chemotherapy (cohort 2; NCT03628716). CV301 is a priming Modified Vaccinia Ankara (MVA) poxvirus followed by boosting fowlpox virus (FPV), which both encode CEA, MUC-1, and the costimulatory molecules IACM-1, LFA-3, and B7-1. Among 32 evaluable patients, two PRs and one CR were seen [137]. The trial was stopped for futility. Three other trials are awaiting results. First, a phase 1 proof-of-concept study was recently completed on muscle-invasive and metastatic UC (NCT03359239). A personalized tumor neoantigen vaccine directed at up to 10 neoantigens identified in a patient’s tumor tissue specimen using a computational neoantigen prediction pipeline was given with atezolizumab [138]. Second, a phase 1/2a trial of intramuscular INO-5401 with INO-9012 followed by electroporation plus atezolizumab is ongoing in patients with mUC after prior ICI or in the front line if cisplatin-ineligible (NCT03502785) [139]. INO-5401 is a mixture of synthetic plasmids encoding WT1, PSMA, and hTERT. INO-9012 is a synthetic plasmid encoding IL-12. Third, an ongoing early phase 1 trial in China is inoculating patients with mUC with the exosomes from the supernatant of stimulated chimeric APC-tumor cells derived from the nuclei of bladder cancer cells and peripheral blood monocytes (NCT05559177). These exosomes will presumably contain tumor antigens and immunostimulatory molecules [140].

Three other trials focus on patients with mUC as one of a few included cancers. The first is a phase 1b trial of a personalized neoantigen mRNA vaccine of up to 20 patient-specific tumor peptides combined with nivolumab for patients with mUC, NSCLC, or melanoma that enrolled 82 patients. The regimen was safe and induced neoantigen-specific immunogenicity (NCT02897765) [141]. Among 15 patients with mUC, the ORR was 27% (95%CI 8–55), including one CR with a mPFS of 5.8 months (95%CI 2.8–12.7) and mOS of 20.7 months (95%CI 4.8-NR). Higher response rates and longer median PFS were seen in melanoma and NSCLC. Second, the personalized neoantigen vaccine of up to 15 patient-specific tumor peptides, NeoPepVac is being studied in combination with an ICI in a phase 1/2 trial for mUC, NSCLC, or melanoma (NCT03715985). Interim results from five patients with melanoma showed initial feasibility, safety, and immunogenicity [142], but final results are pending. Finally, an ongoing multi-arm phase 2 trial is testing the IDO and PD-L1 peptide vaccine IO102-IO103 with pembrolizumab in the front-line for mUC, NSCLC, or head and neck SCC (NCT05077709) [143]. Preliminary results in nine evaluable patients with NSCLC showed a response rate of 44.4% [144]. No results in mUC have yet been reported. This regimen is also being studied in a phase 3 trial in melanoma.

One ongoing single-arm phase 2 trial is studying an oncolytic virus exclusively in patients with mUC in China. This engineered HSV2 oncolytic virus for intratumoral injection is designed to deliver GM-CSF to the tumor microenvironment to potentiate immunotherapy (NCT05248789). Results are awaited.

Other basket trials of vaccines and oncolytic viruses in advanced solid tumors including mUC are summarized in Appendix A.

## 4. Conclusions and Future Directions

The treatment of metastatic urothelial carcinoma currently relies on platinum-based chemotherapy, ICIs, the FGFR inhibitor erdafitinib, and two ADCs, enfortumab vedotin and sacituzumab govitecan. Here, we have summarized the landmark clinical trials that have led to the current standard-of-care use of ICIs as single agents and in combination with EV for mUC. Given the durable responses that have been seen with immunotherapy across cancer types, there is significant excitement in the field of oncology to develop more effective immunotherapeutic regimens. Urothelial carcinoma has proven to be susceptible to immunotherapy and is well poised for future immuno-oncologic developments.

We have summarized recent and ongoing clinical trials investigating ICIs in combination with ADCs, TKIs, and novel antibodies. We expect the results of at least one of these trials to lead to the next change in the standard of care of mUC. In our opinion, the efforts at targeting HER2, combining ICIs with novel ADCs, and triplet regimens are currently among the most promising trials. Further in the future, we may see success with bispecific or trispecific antibodies, though these molecules are still in early-phase clinical development, which is primarily focused on target optimization and proving safety. Thus, finding an effective bispecific antibody for mUC is likely years away. Finally, results from clinical trials of cellular therapies or vaccine-based therapies have the potential to lead to revolutionary breakthroughs for mUC and other solid tumors in the more distant future; however, there are several barriers that need to be overcome before these therapeutic classes integrate into the standard of care. The success of cellular therapies in solid tumors requires the identification of proper targets and costimulatory signaling, clinical planning, and advancements in manufacturing and delivery to make cellular therapeutics logistically feasible, economically viable, and safe.

The results of EV-302 will have a profound impact on the design of current and future clinical trials for mUC. Ongoing trials need to adapt to the new standard of care of frontline pembrolizumab plus EV. Inclusion and exclusion criteria will need to be amended, otherwise trials may have to be terminated in countries where patients have access to pembrolizumab plus EV. For any regimen to be studied in the front-line metastatic setting, the control arm going forward needs to be pembrolizumab plus EV, as it is unethical to randomize patients to platinum-containing chemotherapy given the magnitude of survival benefit of pembrolizumab plus EV compared to platinum-based chemotherapy. Moreover, all second-line-and-beyond trials will need to allow for prior ICI and EV to adequately accrue patients. It is important that the field adapts quickly to the new standard of care to continue the forward momentum of progress.

While the focus of the current review was immunotherapeutic advances in the metastatic setting, there are also ongoing potentially practice-changing immunotherapy studies in the peri-operative setting for muscle-invasive urothelial carcinoma, as well as in the early non-muscle-invasive setting. In these localized disease states, we need to develop treatment approaches that increase cure rates. The current standard of care is neoadjuvant cisplatin-based combination chemotherapy followed by radical cystectomy, then adjuvant nivolumab for patients with pathologic residual disease. Soon we expect adjuvant pembrolizumab to join nivolumab with an FDA approval for this indication based on the press release of the phase 3 Ambassador trial (KEYNOTE-123) that showed a disease-free survival benefit, with ongoing follow-up for OS [145]. There are several ongoing important trials for muscle-invasive disease that incorporate established and novel ICIs as well as circulating tumor DNA minimal residual disease assays for patient selection (ex: NCT05987241), which have been comprehensively reviewed elsewhere [146].

Beyond minimal residual disease biomarkers, there is also a dire need to develop predictive biomarkers for urothelial carcinoma. As there are an increasing number of effective treatment regimens for mUC, the landscape will become increasingly complicated. We ideally will be able to select the regimen to which a patient is most likely to respond and understand optimal sequencing of treatments. Development of predictive biomarkers will require multi-omic biomarker discovery on carefully annotated patient specimens, followed by rigorous external validation and prospective testing. We suspect that for a trial to improve upon the outcomes of pembrolizumab plus EV, predictive biomarker-driven patient selection will be key.

While we await the results of ongoing trials, we need to invest in the next generation of developments. There is a great need to better understand the biology of the cancer–immune response so that the immune system can be intentionally regulated to fight cancer without causing untoward autoimmune toxicities. Novel immunomodulatory molecules have to be discovered. Rational combination regimens must also be efficiently tested in preclinical and then clinical studies for safety and efficacy. Concurrently, cancer- or host-specific predictive biomarkers need to be developed to select the regimen from which a patient is most likely to benefit. Furthermore, as we discover more effective drugs and regimens, we will have to better understand the optimal order of treatment delivery to maximize durable responses and overall survival. The field of immuno-oncology holds great promise to improve outcomes for patients with mUC.

## Figures and Tables

**Figure 2 cancers-16-00335-f002:**
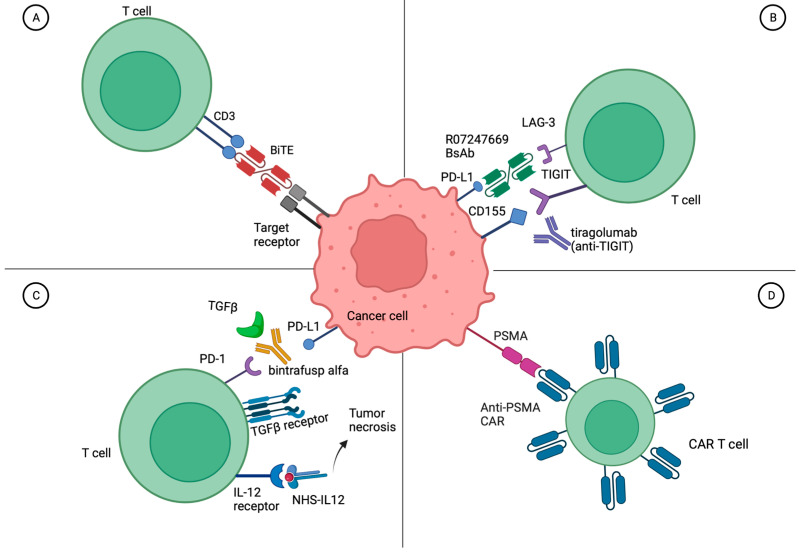
Mechanism of action of bispecific antibodies and cellular therapies in metastatic urothelial carcinoma. (**A**) A representative bispecific T-cell engager (BiTE) binding CD3 on T cells and a target receptor on the tumor cell. (**B**) Tiragolumab (anti-TIGIT) prevents the T-cell inhibitory interaction of CD155 with TIGIT while the bispecific antibody R07247669 binds to LAG-3 on T cells and PD-L1 on cancer cells bringing them into proximity. (**C**) Bintrafusp alfa binds to PD-L1 and simultaneously sequesters immunoinhibitory TGFβ in the tumor microenvironment. Meanwhile, fused NHS-IL12 brings the IL-12 receptor on T cells to areas of tumor necrosis. (**D**) A CAR-T cell engineered to target PSMA (prostate-specific membrane antigen) present on cancer cells. Created with BioRender.

**Table 1 cancers-16-00335-t001:** Clinical trials of novel immune checkpoint inhibitor antibodies.

Regimen	Drug Class	Inclusion Criteria	Phase	Status (March 2023)	Identifier	ORR	mPFS (Months)	mOS (Months)
Retifanlimab	Anti-PD1	mUC, RCC, melanoma, NSCLC	2	Completed	NCT03679767	37.9%	5.7	15.2
Sasanlimab	Anti-PD1	Advanced solid tumors	1	Completed	NCT02573259	21.1%	2.9	10.9
Tislelizumab	Anti-PD1	2L+ mUC after platinum	2	Completed	NCT04004221	24%	2.1	9.8
Rulonilimab	Anti-PD1	1L cis-ineligible or 2L+ mUC	2	Unknown	NCT04636515	NA	NA	NA
Toripalimab	Anti-PD1	1L mUC	2	Unknown	NCT03113266	25.8%	2.3	14.4
Socazolimab	Anti-PDL1	mUC	1/2	Unknown	NCT03676946	NA	NA	NA
Tremelimumab	Anti-CTLA4	mUC after PD1/PDL1	2	Active, not recruiting	NCT03557918	NA	NA	NA

Abbreviations: ORR = objective response rate; mPFS = median progression-free survival; mOS = median overall survival; mUC = metastatic urothelial carcinoma; RCC = renal cell carcinoma, NSCLC = non-small-cell lung carcinoma; 2L+ = second line or beyond; 1L = first line; NA = not available.

**Table 2 cancers-16-00335-t002:** Clinical trials of immune checkpoint inhibitors in the maintenance setting after first-line platinum chemotherapy for metastatic urothelial carcinoma.

Regimen	Drug Classes	Trial Name	Phase	Status (March 2023)	Identifier
Nivolumab + Ipilimumab	Anti-PD1 + Anti-CTLA4	VEXILLUM	2	Recruiting	NCT05219435
Avelumab + Talazoparib	Anti-PDL1 + PARP inhibitor	TALASUR	2	Recruiting	NCT04678362
Avelumab + Lurbinectedin	Anti-PDL1 + chemotherapy		2	Recruiting	NCT05574504
Avelumab + Copanlisib	Anti-PDL1 + PI3K inhibitor		1/2	Not yet recruiting	NCT05687721
Avelumab + MRx0518	Anti-PDL1 + Enterococcus gallinarum	AVENU	2	Withdrawn	NCT05107427
Avelumab + Trilaciclib	Anti-PDL1 + CDK4/6 inhibitor	PRESERVE 3	2	Active, not recruiting	NCT04887831
Avelumab +/− SG or M6223 or NKTR-255	Anti-PDL1 +/− ADC or anti-TIGIT or IL-15 receptor agonist	JAVELIN Bladder Medley	2	Recruiting	NCT05327530
Cohort 4: Cisplatin + SG → maintenance SG + either Avelumab or Zimberelimab Cohort 5: Zimberelimab +/− SG vs. Avelumab	Chemo + ADC then ADC + PD1 or PDL1 Anti-PD1 +/− ADC	TROPHY U-01	2	Recruiting	NCT03547973
Avelumab +/− Cabozantinib	Anti-PDL1 + multi-TKI	MAIN-CAV	3	Recruiting	NCT05092958

Abbreviations: SG = Sacituzumab govitecan; ADC = antibody–drug conjugate; TKI = tyrosine kinase inhibitor.

**Table 3 cancers-16-00335-t003:** Clinical trials of chemotherapy in combination with immune checkpoint inhibitors for metastatic urothelial carcinoma.

Regimen	Drug Classes	Line	Trial Name	Phase	Status (March 2023)	Identifier
Pembrolizumab +/− chemo vs. chemo	Anti-PD1, chemo	1L	KEYNOTE-361	3	Completed	NCT02853305
Atezolizumab +/− chemo vs. chemo	Anti-PDL1, chemo	1L	IMvigor130	3	Active, not recruiting	NCT02807636
Atezolizumab + gemcitabine/cisplatin	Anti-PDL1 + chemo	1L		2	Active, not recruiting	NCT03093922
Atezolizumab + gemcitabine/split-cisplatin	Anti-PDL1 + chemo	1L	AUREA	2	Active, not recruiting	NCT04602078
Nivolumab + chemo vs. ipilimumab/nivolumab vs. chemo	Anti-PD1, anti-CTLA4, chemo	1L	CHECKMATE-901	3	Active, not recruiting	NCT03036098
Ipilimumab + gemcitabine/cisplatin	Anti-CTLA4 + chemo	1L		2	Completed	NCT01524991
Durvalumab + chemo +/− tremelimumab vs. chemo	Anti-PDL1, anti-CTLA4, chemo	1L	NILE	3	Recruiting	NCT03682068
Gemcitabine/cisplatin +/− avelumab	Chemo +/− anti-PDL1	1L	GCISAVE	2	Terminated	NCT03324282
Gemcitabine/platinum +/− tislelizumab	Chemo +/− anti-PD1	1L		3	Recruiting	NCT03967977
Gemcitabine/platinum +/− toripalimab	Chemo +/− anti-PD1	1L		3	Not yet recruiting	NCT04568304
Toripalimab + Nab-paclitaxel +/− cisplatin	Anti-PD1 + chemo	1L		2	Unknown	NCT04211012
Socazolimab + Nab-paclitaxel	Anti-PDL1 + chemo	1L		1	Active, not recruiting	NCT04603846
Gemcitabine/carboplatin +/− avelumab	Chemo +/− anti-PDL1	1L	INDUCOMAIN	2	Completed	NCT03390595
Nivolumab + gemcitabine + either carboplatin or oxaliplatin	Anti-PD1 + chemo	1L		2	Active, not recruiting	NCT03451331
Pembrolizumab + either docetaxel or gemcitabine	Anti-PD1 + chemo	2L+		1b/2	Completed	NCT02437370
Avelumab + docetaxel	Anti-PDL1 + chemo	2L+	AVETAX	1b	Active, not recruiting	NCT03575013
Pembrolizumab + nab-paclitaxel	Anti-PD1 + chemo	2L+	PEANUT	2	Completed	NCT03464734
Pembrolizumab + paclitaxel	Anti-PD1 + chemo	2L+		2	Completed	NCT02581982
Atezolizumab +/− eribulin	Anti-PDL1 +/− chemo			2	Active, not recruiting	NCT03237780
Anti-PD1 + liposomal doxorubicin	Anti-PD1 + chemo	2L+		2	Unknown	NCT04101812
Avelumab + pemetrexed in MTAP-deficient	Anti-PDL1 + chemo	2L+		2	Active, not recruiting	NCT03744793
Zimberelimab + pemetrexed + etrumadenant in MTAP-deficient	Anti-PD1 + chemo + adenosine receptor antagonist	2L+		2	Recruiting	NCT05335941

Abbreviations: 1L = first line; 2L = second line or beyond.

**Table 5 cancers-16-00335-t005:** Clinical trials of tyrosine kinase inhibitors in combination with immune checkpoint inhibitors for metastatic urothelial carcinoma.

Regimen	Drug Classes	Inclusion Criteria	Trial	Phase	Status (March 2023)	Identifier	ORR	mPFS (Months)	mOS (Months)
Erdafitinib +/− cetrelimab +/− platinum chemo	FGFRi +/− anti-PD1 +/− chemo	mUC with FGFR2/3 alterationPhase 2: 1L cisplatin-ineligible mUC	NORSE	1b/2	Active, not recruiting	NCT03473743	54.5%	10.97	NA
Futibatinib + pembrolizumab	FGFRi + anti-PD1	Platinum-ineligible mUC		2	Recruiting	NCT04601857	NA	NA	NA
Rogaratinib + atezolizumab vs. atezolizumab	FGFRi + anti-PDL1 vs. anti-PDL1	1L mUC cisplatin-ineligible, FGFR1/3 (+) by RNAscope	FORT-2	1b/2	Active, not recruiting	NCT03473756	54%	NA	NA
Derazantinib +/− atezolizumab	FGFR/CSF1R inhibitor +/− anti-PDL1	2L+ mUC, 1L cisplatin-ineligible mUC	FIDES-02	1b/2	Completed	NCT04045613	NA	NA	NA
LOXO-435 +/− pembrolizumab	FGFR3i +/− anti-PD1	FGFR3-altered advanced solid tumors including mUC		1	Recruiting	NCT05614739	NA	NA	NA
Pemigatinib +/− pembrolizumab vs. gemcitabine + carboplatin	FGFRi +/− anti-PD1 vs. chemo	1L cisplatin-ineligible mUC with an FGFR3 mutation or rearrangement	FIGHT-205	2	Terminated	NCT04003610	NA	NA	NA
Cabozantinib + nivolumab +/− ipilimumab	Multi-TKI + anti-PD1 +/− anti-CTLA4	Metastatic GU cancers including mUC		1	Active, not recruiting	NCT02496208	38.5%, 16.0%	12.8	25.4
Cabozantinib + nivolumab + ipilimumab	Multi-TKI + anti-PD1 + anti-CTLA4	Rare GU tumors, metastatic bladder cancer histologic variants	ICONIC	2	Recruiting	NCT03866382	NA	NA	NA
Cabozantinib + durvalumab	Multi-TKI + anti-PDL1	mUC and non-UC histologies	ARCADIA	2	Unknown	NCT03824691	39.7%	7.6	11.6
Cabozantinib + pembrolizumab	Multi-TKI + anti-PD1	1L cisplatin-ineligible mUC	PemCab	2	Active, not recruiting	NCT03534804	NA	NA	NA
Lenvatinib + pembrolizumab vs. placebo + pembrolizumab	Multi-TKI + anti-PD1	1L cisplatin-ineligible PD-L1 (-) or platinum- ineligible mUC	LEAP-011	3	Active, not recruiting	NCT03898180	NA	HR 0.91 (0.71–1.16)	HR 1.25 (0.94–1.67)
Sitravatinib + nivolumab	Multi-TKI + anti-PD1	8 cohorts of mUC		2	Terminated	NCT03606174	0–33%	3.5–7.8	NA
Famitinib + camrelizumab	Multi-TKI + anti-PD1	Advanced GU and gynecologic cancers		2	Recruiting	NCT03827837	30.6%	4.1	12.9
XL092 +/− nivolumab +/− either ipilimumab or relatlimab	Multi-TKI +/− anti-PD1 +/− anti-CTLA5 or anti-LAG3	Advanced solid tumors including mUC	STELLAR-002	1b	Recruiting	NCT05176483	NA	NA	NA

Abbreviations: FGFRi = fibroblast growth factor receptor inhibitor; mUC = metastatic urothelial carcinoma; ORR = objective response rate; mPFS = median progression-free survival; mOS = median overall survival; 1L = first line; 2L+ = second line or beyond; NA = not available; HR = hazard ratio.

**Table 6 cancers-16-00335-t006:** Clinical trials of novel antibodies alone and in combination with immune checkpoint inhibitors for metastatic urothelial carcinoma.

Regimen	Drug Classes	Inclusion Criteria	Phase	Status (March 2023)	Identifier
INCAGN01949 +/− nivolumab +/− ipilimumab	Anti-OX40 Ab +/− anti-PD1 +/− anti-CTLA4	Advanced solid tumors	1/2	Completed	NCT03241173
Atezolizumab +/− vonlerolizumab	Anti-PDL1 +/− anti-OX40 Ab	1L mUC cisplatin-ineligible	2	Terminated for slow accrual	NCT03029832
BGB-A445 +/− tislelizumab	Anti-OX40 Ab +/− anti-PD1	2L+ metastatic UC, RCC, melanoma	1b/2	Recruiting	NCT05661955
ICAGN01876 +/− nivolumab +/− ipilimumab	Anti-GITR Ab +/− anti-PD1 +/− anti-CTLA4	Advanced solid tumors	1/2	Completed	NCT03126110
LYT-200 +/− tislelizumab or gemcitabine + nab-paclitaxel	Anti-α-GAL-9 +/− anti-PD1 +/− chemo	Advanced solid tumors	1/2	Recruiting	NCT04666688
ABBV-151 + budigalimab	Anti-GARP-TGFβ1 + anti-PD1	Advanced solid tumors	1	Recruiting	NCT03821935
NGM831 +/− pembrolizumab	Anti-ILT3 +/− anti-PD1	Advanced solid tumors	1/1b	Recruiting	NCT05215574
NGM438 +/− pembrolizumab	Anti-LAIR1 +/− anti-PD1	Advanced solid tumors	1/1b	Recruiting	NCT05311618
AZD8853	Anti-MIC-1	2L+ mUC, NSCLC, colorectal	1/2a	Terminated	NCT05397171

Abbreviations: mUC = metastatic urothelial carcinoma; UC = urothelial carcinoma; RCC = renal cell carcinoma; NSCLC = non-small-cell lung cancer; 1L = first line; 2L+ = second line or beyond.

**Table 7 cancers-16-00335-t007:** Clinical trials of cellular therapies for metastatic urothelial carcinoma.

Regimen	Drug Classes	Target	Inclusion Criteria	Phase	Status (March 2023)	Identifier
WT-1 dendritic cells	Autologous dendritic cells	WT-1	mUC	1	Completed	Japan (UMIN 000027279) [125]
TILs	Autologous TILs		mUC after platinum, ICI	2	Unknown	NCT04383067
LN-145 + pembrolizumab	Autologous TILs + anti-PD1		mUC after cisplatin	2	Withdrawn (no accrual)	NCT03935347
Autologous central memory T cells	Autologous central memory T cells		mUC after 1L gemcitabine + cisplatin	2	Terminated (shortage of funds)	NCT03389438
4SCAR-T cells	CAR-T cells	PSMA + FRα	mUC without SOC option	1/2	Unknown	NCT03185468
PD-1 knockout engineered T cells	Autologous engineered T cells	PD-1 knockout	mUC	1	Withdrawn (no funding)	NCT02863913
Autologous cytokine-induced killer cells	Autologous NK cells		mUC, metastatic RCC	1/2	Not yet recruiting	NCT05108077

Abbreviations: TILs = tumor-infiltrating lymphocytes; NK = natural killer; PSMA = prostate-specific membrane antigen; mUC = metastatic urothelial carcinoma; IC = immune checkpoint inhibitor; 1L = first line; SOC = standard of care; RCC = renal cell carcinoma.

**Table 8 cancers-16-00335-t008:** Clinical trials of vaccines for metastatic urothelial carcinoma.

Regimen	Drug Classes	Target	Inclusion Criteria	Phase	Status (March 2023)	Identifier	ORR	mPFS (Months)	mOS (Months)
PGV001 + atezolizumab	Personalized tumor neoantigen vaccine + anti-PDL1		mUC	1	Completed	NCT03359239	NA	NA	NA
CV301 + atezolizumab	Poxviruses encoding CEA, MUC-1 + anti-PDL1	CEA, MUC-1	mUC	2	Completed	NCT03628716	9.4%	NA	NA
INO-5401 + INO-9012 + atezolizumab	Plasmids encoding WT1, PSMA, hTERT, IL12 + anti-PDL1	WT1, PSMA, hTERT	mUC	1/2a	Active, not recruiting	NCT03502785	NA	NA	NA
Chimerical exosomal tumor vaccine	Exosomes in supernatant of chimeric APC-tumor cells		mUC	1	Recruiting	NCT05559177	NA	NA	NA
NEO-PV-01 + nivolumab	Personalized tumor neoantigen mRNA vaccine + anti-PD1		mUC, NSCLC, melanoma	1b	Completed	NCT02897765	27%	5.8	20.7
NeoPepVac EVAX-01-CAF09b + anti-PD1/PDL1	Personalized tumor neoantigen vaccine		mUC, NSCLC, melanoma	1/2	Active, not recruiting	NCT03715985	NA	NA	NA
IO102-IO103 + pembrolizumab	IDO + PD-L1 peptide vaccine + anti-PD1	IDO, PD-L1	1L mUC, NSCLC, SCCHN	2	Recruiting	NCT05077709	NA	NA	NA
OH2	Oncolytic virus delivering GM-CSF gene		mUC	2	Recruiting	NCT05248789	NA	NA	NA

Abbreviations: ORR = objective response rate; mPFS = median progression-free survival; mOS = median overall survival; mUC = metastatic urothelial carcinoma; mRNA = messenger RNA; NA = not available; GM-CSF = granulocyte-macrophage colony-stimulating factor.

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
