# Peer review of "A Comprehensive Review of Immunotherapy Clinical Trials for Metastatic Urothelial Carcinoma: Immune Checkpoint Inhibitors Alone or in Combination, Novel Antibodies, Cellular Therapies, and Vaccines"

_cancers, 2024, doi:10.3390/cancers16020335_

Round 1

Reviewer 1 Report

Comments and Suggestions for Authors

In this comprehensive review the authors discuss the current landscape of immunotherapy trials for advanced urothelial cancer. Overall the manuscript is well written in clear English it is easy to understand. The scope of the manuscript is very broad covering everything from immune checkpoint inhibitors in combination with chemotherapy and targeted therapy to vaccines, and cell therapies. Overall, this is a well executed compilation of studies from literature. However, there are some important limitations that should be addressed as below.

Major:

In general, this is effectively a catalog of studies. While the cataloging of the landscape of immunotherapy treatment for metastatic urothelial carcinoma is important, the authors provide little to no expert perspective on the topics discussed or critiques of the trials referenced (i.e. did patients receive adequate post-protocol therapy in randomized phase 3 studies?, how do eligibility criteria compare between studied and which are most reflective of patients in a community setting?, etc) . The manuscript would benefit greatly from contextualization of each section in the broader cancer therapy landscape and historical treatment landscape of urothelial carcinoma. The discussion at the end could also be expanded to the same effect.

One challenge with getting into the phase 1 studies (i.e. bispecifics, cell therapies, vaccines) is that there are hundreds of trials that include UC, so any review of studies at this phase inevitably omit some where UC is a focus, and include others where UC has been deprioritized. This can't be known without direct involvement in the studies, so it may make sense to focus only on published phase 1 immunotherapy studies with a UC focus. 

The treatment landscape has now changed with the presentation of more data from EV-302 at ESMO in 2023. The manuscript should be updated in several sections to reflect these new transformative data in urothelial cancer. Checkmate 901 is now published as well. 

Minor:

Introduction:

Line 44 - the first paragraph would read more clearly if the first sentence was omitted.

Section 1

Line 95 - not all ICI approvals are independent of PD-L1 status. Pemrbolizumab has an FDA approval for PD-L1+ cisplatin-ineligible UC.

Reviewer 2 Report

Comments and Suggestions for Authors

1. Please update with data presented at ESMO congress 2023- especially EV302, CHECKMATE901 and THOR.

2. The Ambassador Phase III adjuvant pembrolizumab trial was reported in a press release to be positive.

3. Relatlimab is being planned to be evaluated in the adjuvant setting in the MODERN trial.

4. The potential role of ctDNA in assessing MRD and guiding precision medicine could be highlighted.
